

# Metagenomics of African *Empogona* and *Tricalysia* (Rubiaceae) reveals the presence of leaf endophytes

Brecht Verstraete[1,*], Steven Janssens[1,2,*], Petra De Block[1], Pieter Asselman[3], Gabriela Méndez[4,5], Serigne Ly[6], Perla Hamon[6] and Romain Guyot[6,7]

[1] Meise Botanic Garden, Meise, Belgium
[2] Department of Biology, KU Leuven, Leuven, Belgium
[3] Department of Biology, Ghent University, Ghent, Belgium
[4] Grupo de Investigación (BIOARN), Universidad Politécnica Salesiana, Quito, Ecuador
[5] Facultad de ingenieria, Pontificia Universidad Católica del Ecuador, Quito, Ecuador
[6] DIADE, Université de Montpellier, Montpellier, France
[7] Department of Electronics and Automation, Universidad Autónoma de Manizales, Manizales, Colombia
[*] These authors contributed equally to this work.

Corresponding author
Brecht Verstraete,
brecht.verstraete@plantentuinmeise.be

## ABSTRACT

**Background.** Leaf symbiosis is a phenomenon in which host plants of Rubiaceae interact with bacterial endophytes within their leaves. To date, it has been found in around 650 species belonging to eight genera in four tribes; however, the true extent in Rubiaceae remains unknown. Our aim is to investigate the possible occurrence of leaf endophytes in the African plant genera *Empogona* and *Tricalysia* and, if present, to establish their identity.

**Methods.** Total DNA was extracted from the leaves of four species of the Coffeeae tribe (*Empogona congesta, Tricalysia hensii, T. lasiodelphys,* and *T. semidecidua*) and sequenced. Bacterial reads were filtered out and assembled. Phylogenetic analysis of the endophytes was used to reveal their identity and their relationship with known symbionts.

**Results.** All four species have non-nodulated leaf endophytes, which are identified as *Caballeronia*. The endophytes are distinct from each other but related to other nodulated and non-nodulated endophytes. An apparent phylogenetic or geographic pattern appears to be absent in endophytes or host plants. *Caballeronia* endophytes are present in the leaves of *Empogona* and *Tricalysia*, two genera not previously implicated in leaf symbiosis. This interaction is likely to be more widespread, and future discoveries are inevitable.

## INTRODUCTION

Plant-bacteria interactions are considered ubiquitous and a common phenomenon in angiosperms (*Orozco-Mosqueda & Santoyo, 2021*). Many studies have shown the beneficial impact of such interactions; the most widely known example is the nitrogen-fixing *Rhizobia*

bacteria that occur in the rhizosphere of several members of the Fabaceae family (*Poole, Ramachandran & Terpolilli, 2018*).

An example of plant endophytes in the phyllosphere is bacterial leaf nodule symbiosis, which is found in a number of taxa in eudicots (Primulaceae and Rubiaceae) and monocots (Dioscoreaceae) (*Miller, 1990*). When leaf nodules are present, it is usually easy to recognize the presence of this particular plant-bacteria interaction. Moreover, these distinctive structures can sometimes be used for the taxonomic characterisation of certain plant lineages in Rubiaceae (*e.g., Van Oevelen et al., 2001*; *Razafimandimbison et al., 2017*). However, leaf nodules are not necessarily always present, and therefore putative leaf endophytes often remain undetected (*Verstraete et al., 2011*; *Lemaire et al., 2012b*). With the development of modern molecular methods and especially with the rapid increase of different high-throughput sequencing techniques, an array of tools became available to detect and study bacterial leaf endophytes (*e.g., Carlier et al., 2016*; *Carlier et al., 2017*; *Danneels et al., 2021*; *Schindler et al., 2021*; *Danneels et al., 2023*).

The Rubiaceae family currently has the highest recorded number of species that are characterized by bacterial leaf nodulation. The presence of "thickened, hard warts" in *Pavetta indica* L. was already noted almost 130 years ago (*Trimen, 1894*) and it was later discovered that these leaf nodules contain endophytic bacteria (*Zimmermann, 1902*). This symbiosis between Rubiaceae and leaf bacteria was then more elaborately described by *Von Faber (1912)*. Currently, leaf nodules in Rubiaceae have been observed in the genera *Pavetta* L. (ca 350 spp in the Pavetteae tribe), *Psychotria* L. (ca 80 spp in the Psychotrieae tribe), and *Sericanthe* Robbr. (ca 12 spp in the Coffeeae tribe) (*Lersten & Horner, 1976*; *Miller, 1990*; *Lemaire et al., 2011b*; *Lemaire et al., 2012a*). Because the leaf bacteria cannot survive outside the nodules, culture-independent methods were necessary to establish the identity of these nodulated endophytes: they belong to the genus *Burkholderia* s.l. (*e.g., Van Oevelen et al., 2002*; *Lemaire et al., 2011a*; *Lemaire et al., 2012a*; *Pinto-Carbó et al., 2018*). Furthermore, most plant species seem to harbour unique bacterial lineages. Since the discovery of leaf endophytes, the *Burkholderia* s.l. genus has undergone several taxonomic changes and therefore names such as *Paraburkholderia* and *Caballeronia* can also be encountered in the literature (*Bach et al., 2022*).

The Rubiaceae family also contains species with leaf endophytes that are not housed in conspicuous nodules (*Van Wyk et al., 1990*). Instead, this second phenotype is characterised by endophytes occurring in the intercellular space between the leaf mesophyll cells (*Van Wyk et al., 1990*; *Lemaire et al., 2012b*; *Verstraete et al., 2013a*)). This non-nodulating phenotype has been observed in the genus *Psychotria* (22 spp in the Psychotrieae tribe; *Lemaire et al., 2012b*) as well as in the genera *Fadogia* Schweinf., *Fadogiella* Robyns, *Globulostylis* Wernham, *Rytigynia* Blume, and *Vangueria* Juss. (ca 191 spp in the Vanguerieae tribe; *Verstraete et al., 2011*; *Verstraete et al., 2013a*; *Verstraete et al., 2013b*). These non-nodulated endophytes have also been identified as *Burkholderia* s.l. but they are not necessarily specific to a single host plant species (*Lemaire et al., 2012b*; *Verstraete et al., 2011*; *Verstraete et al., 2013a*; *Verstraete et al., 2013b*). Because the same bacterial species can be found in several host species or even in the soil, this interaction is believed to be less specialised (*Verstraete et al., 2013a*; *Verstraete et al., 2014*).

We currently know that leaf symbiosis (both nodulated and non-nodulated) is present in eight genera in four tribes of Rubiaceae but the true extent remains unknown to date. However, it is likely that leaf symbiosis is more widespread and could occur in other genera as well. *Tilney & Van Wyk (2009)* made histological sections of leaves of *Keetia gueinzii* (Sond.) Bridson (Vanguerieae tribe) and saw "intercellular, non-nodulating, slime-producing bacteria", but this has not been confirmed with molecular data yet. Several *Burkholderia* species have been found to be associated with the roots of coffee plants (*i.e., Coffea arabica* L. and *C. canephora* Pierre ex A.Froehner in *Caballero-Mellado et al. (2004)*, and *C. liberica* W.Bull in *Duong et al., 2021*) or with the seeds (see review of *Vaughan, Mitchell & Mc Spadden Gardener, 2015*). *Burkholderia* bacteria have also been found to be associated with the leaves of *C. arabica*, although only as epiphytes and not as endophytes (*Vega et al., 2005*).

The genus *Coffea* belongs to the Coffeeae tribe together with the genus *Sericanthe*, for which leaf nodules have been reported (*Lemaire et al., 2011a*). So far, there have been no reports of other taxa in this tribe that contain *Burkholderia* s.l. endophytes, not in leaf nodules nor free between the mesophyll cells. The genera within the Coffeeae tribe that are most closely related to *Sericanthe* are *Diplospora* DC., *Empogona* Hook.f., and *Tricalysia* A.Rich ex DC (*Arriola et al., 2018*). Since none of these genera have visible nodules in their leaves, if leaf endophytes were to be present, it would have to be non-nodulated ones. Non-nodulated leaf endophytes in Rubiaceae have so far only been found in taxa occurring in Africa (*Lemaire et al., 2012b*; *Verstraete et al., 2013b*). The genus *Diplospora* occurs in (sub)tropical Asia, while the other three genera (*Empogona*, *Sericanthe*, and *Tricalysia*) are found in continental Africa and Madagascar (*POWO, 2023*). It is plausible that the highest likelihood of finding additional taxa with leaf endophytes are taxa closely related to *Sericanthe* and occurring in Africa, and we therefore focus our efforts on *Empogona* and *Tricalysia*.

Our specific aims are (1) to investigate the possible occurrence of non-nodulated leaf endophytes in *Empogona* and *Tricalysia*, (2) if present, to establish their identity and to explore their phylogenetic relationships with other nodulated and non-nodulated endophytes, and (3) to look for patterns in the endophytes and host plants.

## MATERIALS & METHODS

### Plant material, DNA isolation, and sequencing

This study investigates four species of the Coffeeae tribe: *Empogona congesta* (Oliv.) J.E.Burrows, *Tricalysia hensii* De Wild., *T. lasiodelphys* (K.Schum. & K.Krause) A.Chev., and *T. semidecidua* Bridson (Table 1). These species were included in a previous study about chloroplast genome evolution in Rubiaceae (*Ly et al., 2020*). The plant material was obtained from the collection of Meise Botanic Garden, Belgium. Total DNA isolation from the leaves and DNA sequencing was done as described in *Ly et al. (2020)*. Raw Illumina reads (BGI-seq 500 platform, $2 \times 100$ bp paired-end) are available under the BioProject PRJNA880288 (Table 1). While processing the raw sequencing reads, *Ly et al. (2020)* encountered "contamination" (*i.e.,* non-chloroplast reads) and removed those reads to be

 

**Table 1** Provenance of the material of the four investigated plant species, deposited at Meise Botanic Garden (https://www.botanicalcollections.be), and information about the raw sequencing reads obtained from the total DNA isolated.

| Species name | Barcode of voucher | Country | Number of reads | Number of nucleotides (Gb) | NCBI accession number |
|---|---|---|---|---|---|
| *Empogona congesta* | BR6202001591004 | Zambia | 2 × 65,763,918 | 13.15 | SRR21547710 |
| *Tricalysia hensii* | BR0000012568055 | D.R. Congo | 2 × 61,592,477 | 12.31 | SRR21547709 |
| *Tricalysia lasiodelphys* | BR0000009955950 | Cameroon | 2 × 62,973,455 | 12.59 | SRR21547708 |
| *Tricalysia semidecidua* | BR6202001590007 | Zambia | 2 × 65,171,012 | 13.03 | SRR21547707 |

able to reconstruct the chloroplast genomes of these four species. However, in this study, we are interested in this "contamination" in the raw reads because it contains information on possible leaf endophytes.

## Read filtering and assembly

The contaminants in the Illumina reads were explored using metagenomic analysis tools. First, Kaiju v.1.9 (*Menzel, Ng & Krogh, 2016*) was used to identify and classify raw reads not belonging to the plant genome with the NCBI non-redundant RefSeq protein database (NCBI nr_euk). When a significant number of contaminants was present, it was examined in more detail in the Kaiju output. Second, the presence of contaminants was confirmed by exploring the distribution of GC count per sample using FASTQC: several GC count peaks in a single sample might suggest the presence of different organisms in the reads. Finally, reads from contaminants showing different GC count were filtered out using KAT v.2.3.4 (k-mer Analysis Tool; *Mapleson et al., 2017*). The reads were analysed with KAT gcp to create a matrix of the number of k-mers found, given k-mer frequency (27-mer) with GC count for each distinct k-mer to explore GC bias. The matrix was displayed *via* a density plot of the k-mer coverage *versus* GC count. KAT filter tools were used to filter out reads according to the GC bias (k-mer coverage of 100 to 500X and GC count of 10 to 22%). The reads left after filtering (without trimming, cleaning, or error correction) were subsequently assembled using MaSuRCA v.3.2.6 (*Zimin et al., 2013*) into scaffolds (Table 2) with the default parameters. The draft genome assemblies of the four endophytes are available on GenBank (Table 3) and on Zenodo (*Verstraete et al., 2022a*). Additionally, the reads were assembled using metaSPAdes v.3.15.5 (*Nurk et al., 2017*) and those draft assemblies are also available on Zenodo (*Verstraete et al., 2023*).

## Analysis of the assembled bacterial draft genomes

The scaffolds obtained after assembly were compared to a *Burkholderia* s.l. sequence database of 2,288 genomes (representing 22 Gb) downloaded from NCBI (https://www.ncbi.nlm.nih.gov/assembly/?term=burkholderia) as available in September 2019. BLASTn v.2.2.26 (NCBI BLAST) was used for the comparison. Scaffolds with an e-value <10 $e^{-20}$ were kept. Assembly completeness was assessed using BUSCO v.5.4.3 (*Seppey, Manni & Zdobnov, 2019*) with the proteobacteria_odb10 database downloaded from https://busco.ezlab.org. The assembled draft genomes were annotated using Prokka v.1.14.5 (*Seemann, 2014*) and the annotations are available on Zenodo (*Verstraete et al., 2023*).

**Table 2  Statistics on the filtered and assembled bacterial reads in *Empogona* and *Tricalysia*.**

| Host species | Number of filtered reads (100 bp) | Number of scaffolds | Assembly N50 (bp) | Assembly size (Mb) | Average coverage |
|---|---|---|---|---|---|
| *Empogona congesta* | 2 × 14,051,241 | 632 | 9,820 | 3.863 | 727X |
| *Tricalysia hensii* | 2 × 11,799,331 | 736 | 48,029 | 7.818 | 301X |
| *Tricalysia lasiodelphys* | 2 × 9,204,217 | 2,971 | 2,095 | 4.340 | 424X |
| *Tricalysia semidecidua* | 2 × 23,251,991 | 1,578 | 22,085 | 4.082 | 1139X |

**Table 3  Statistics about the scaffolds after BLASTn filtering against *Burkholderia* s.l. genomes.**

| Host species | Number of scaffolds with BLASTn hits against *Burkholderia* s.l. genomes (e-value 10 e$^{-20}$) | Assembly N50 of filtered scaffolds (bp) | Assembly size of filtered scaffolds (Mb) | Complete BUSCO scores | Missing BUSCO scores | NCBI accession number |
|---|---|---|---|---|---|---|
| *Empogona congesta* | 612 | 10,140 | 3.762 | 93.2% | 1.8% | JAQFVJ000000000 |
| *Tricalysia hensii* | 369 | 60,447 | 6.951 | 99.6% | 0.4% | JAQFVG000000000 |
| *Tricalysia lasiodelphys* | 2,644 | 2,165 | 4.145 | 75.3% | 6.4% | JAQFVH000000000 |
| *Tricalysia semidecidua* | 490 | 24,168 | 3.919 | 95.4% | 1.4% | JAQFVI000000000 |

## Assembly of the 16S rRNA genes

A custom pipeline was developed to assemble targeted genes from the raw Illumina reads (Mendez Silva et al., unpublished). In short, raw Illumina reads were mapped to a 16S rRNA reference gene (CP000010.1: c2677815–2676328 *Burkholderia mallei*) using Bowtie2 v.2.4.4 (*Langmead & Salzberg, 2012*). Mapped reads were subsequently filtered out and assembled with ABySS v.2.2.1 (*Jackman et al., 2017*).

## Phylogenetic analysis of the endophytes

Three genes (16 rRNA, *gyrB*, and *recA*) were identified and used for phylogenetic analysis. A FASTA file with these sequences is available on Zenodo (*Verstraete et al., 2022b*). The 16S rRNA sequences were obtained from the raw Illumina reads, while the *gyrB* and *recA* sequences were obtained from the assembled scaffolds using the BLASTn tool. The *gyrB* nucleotide sequence (NC_006348.1: 3081–5549 *Burkholderia mallei*) and the *recA* nucleotide sequence (NC_006348.1: 290127–291197 *Burkholderia mallei*) were used as queries. Sequences were extracted using the extractseq function as implemented in EMBOSS (*Rice, Longden & Bleasby, 2000*).

The obtained sequences were combined with previously published datasets (*Lemaire et al., 2011b*; *Lemaire et al., 2012b*; *Verstraete et al., 2013b*; *Danneels et al., 2023*) to assess the phylogenetic position of the detected endophytes (Table S1). Automatic sequence alignment was performed with MAFFT v.7.490 (*Katoh et al., 2002*), followed by manual optimisation in Geneious R11. Possible incongruence among the different datasets was tested using a partition homogeneity test (implemented in PAUP*4.0b10a; *Swofford, 2003*). Due to sensitivity issues with the latter test (*Barker & Lutzoni, 2002*), resolution and support values of the different topologies were visually examined (hard *versus* soft incongruence;

*Johnson & Soltis, 1998*). The best-fit nucleotide substitution model for each gene marker was selected using the Akaike information criterion in jModelTest v.2.1.10 (*Posada, 2008*). The model selection test showed that the GTR+I+G model is the most optimal model for 16S rRNA, and that the GTR+G model is the best for *gyrB* and *recA*. Bayesian inference analyses were performed with MrBayes v.3.2.7 (*Huelsenbeck & Ronquist, 2001*) on three individual data partitions and a combined data matrix under a mixed-model approach. Ten million generations were run, and parameters and trees were sampled every 1,000th generation. Chain convergence and ESS parameters were checked with Tracer v.1.7.2 (*Rambaut et al., 2018*). Bayesian posterior probability values above or equal to 0.95 are regarded as statistically supported (*Alfaro, Zoller & Lutzoni, 2003*).

## RESULTS

### Read filtering and assembly

Several species of Rubiaceae were recently sequenced using a whole genome sequencing approach with DNA extracted from leaves to establish their phylogenetic relationships (*Ly et al., 2020*). During the quality analysis steps, two types of raw reads with very different GC count per read (one peak at 39% and a second at 63%) were found, suggesting the probable presence of multiple organisms in the raw reads. To understand the origin of the different GC count peaks, a metagenomic approach was applied on the raw reads from *Empogona* and *Tricalysia* species.

For *E. congesta*, 37% of all reads could be assigned to a taxon name based on the NCBI non-redundant RefSeq protein database. About 30% of all reads or 80% of the named reads is assigned to Burkholderiaceae (Fig. S1A). For *T. semidecidua*, 30% of all reads is assigned to a taxon name and about 25% of all reads or 83% of the named reads is assigned to Burkholderiaceae (Fig. S1C). For *T. lasiodelphys*, 23% of all reads is assigned to a taxon name and about 18% of all reads or 75% of the named reads is assigned to Burkholderiaceae (Fig. S1E). For *T. hensii*, 20% of all reads is assigned to a taxon name and about 14% of all reads or 73% of the named reads is assigned to Burkholderiaceae (Fig. S1G). The Kaiju output can also be found in Data S1.

To confirm the presence of raw reads belonging to bacteria, the levels of k-mer coverage and GC count per distinct k-mer were analysed. The density plots of the k-mer coverage and the GC count indicate a low to medium k-mer coverage with a wide spread of the GC count (Figs. S1B, S1D, S1F and S1H, left part of the plots) suggesting the presence of reads with sequencing error and a medium coverage genome (*i.e.,* the plant genome). However, the plots also show high coverage (approximately 700X) with GC counts of 15 to 20% (Fig. S1B, S1D, S1F and S1H, upper right of the plots). These unexpected reads with high coverage and high GC count were extracted from the set of raw reads using KAT.

The filtered bacterial reads from *E. congesta* were assembled into 632 scaffolds with an assembly size of 3.8 Mb, while the bacterial reads from *T. hensii, T. lasiodelphys*, and *T. semidecidua* were assembled into 769, 736, and 1,578 scaffolds with assembly sizes of 5.7, 7.8, and 4 Mb, respectively. The assembly N50 ranged from 9.8 to 48 Kb (Table 2).

## Analysis of the assembled bacterial draft genomes

For the host species *E. congesta,* 612 of the 632 scaffolds (97%) resulted in a strong hit against the *Burkholderia* s.l. database (e-value cut-off 10 e$^{-20}$). For the draft genomes of the host species *T. lasiodelphys, T. hensii*, and *T. semidecidua*, there was a lower proportion of hits (89%, 50%, and 31%, respectively) (Table 3). Removing scaffolds without a strong hit against the *Burkholderia* s.l. genome database increased the N50 of the assemblies. However, this did not result in a large decrease in assembly sizes, indicating that only small-size scaffolds were discarded. BUSCO analysis indicated high completeness of the assemblies (>90%), except for the draft genome of the host species *T. lasiodelphys* (75.3%). In contrast to the other species, the large number of scaffolds and low N50 value for *T. lasiodelphys* suggest a fragmented and incomplete assembly. The genome assemblies should be considered as rough drafts, since bias might have been introduced when filtering the sequences by k-mer/GC count and BLAST. However, it is unlikely that host plant sequences remain present in the assemblies.

## Phylogenetic analysis of leaf endophytes in *Empogona* and *Tricalysia*

The phylogenetic position of the endophytes found in *Empogona congesta*, *Tricalysia hensii*, *T. lasiodelphys*, and *T. semidecidua* were inferred from the 16S rRNA, *gyrB*, and *recA* sequences. The combined dataset demonstrated that all four non-nodulating leaf endophytes of *Empogona* and *Tricalysia* belong to *Burkholderia* s.l., more specifically, to the genus *Caballeronia* (Fig. 1). The non-nodulated endophyte of *T. lasiodelphys* is related to the nodulated endophytes of *Sericanthe andongensis* (Hiern) Robbr. and the non-nodulated endophytes of *Psychotria psychotrioides* (DC.) Roberty (BPP: 0.63). The non-nodulated endophyte of *E. congesta* falls within a highly supported clade of nodulated endophytes of several *Pavetta* species and non-nodulated endophytes of several *Globulostylis* species (BPP: 1.00). The non-nodulated endophyte of *T. hensii* is found as sister to the nodulated *Candidatus* Burkholderia kikwitensis (BPP: 1.00), nested within a clade of several other nodulated endophytes of *Psychotria* species. The non-nodulated endophyte of *T. semidecidua* falls in a clade with *Caballeronia fortuita* and *C. novacaledonica* (BPP: 0.97).

# DISCUSSION

## Detection of non-nodulated leaf endophytes in *Empogona* and *Tricalysia*

The majority of the studies on phylogenetic relationships within the Rubiaceae family to date has relied on phylogenetic approaches using individual nuclear or plastid DNA markers, or a combination of both (*Wikström, Bremer & Rydin, 2020*). However, phylogenomic approaches using more comprehensive amounts of genetic data are becoming more and more common, *e.g.*, mitochondrial genomic data (*Rydin, Wikström & Bremer, 2017*), plastid genomes (*Ly et al., 2020*; *Wikström, Bremer & Rydin, 2020*), or a combination of hundreds of nuclear genes (*Antonelli et al., 2021*; *Thureborn et al., 2022*). The onset of the high-throughput sequencing methodology provides a novel tool to also detect leaf endophytes in Rubiaceae. High-throughput sequencing allows for the sequencing of total DNA, which is subsequently cleaned using bioinformatic filtering to only retain desired
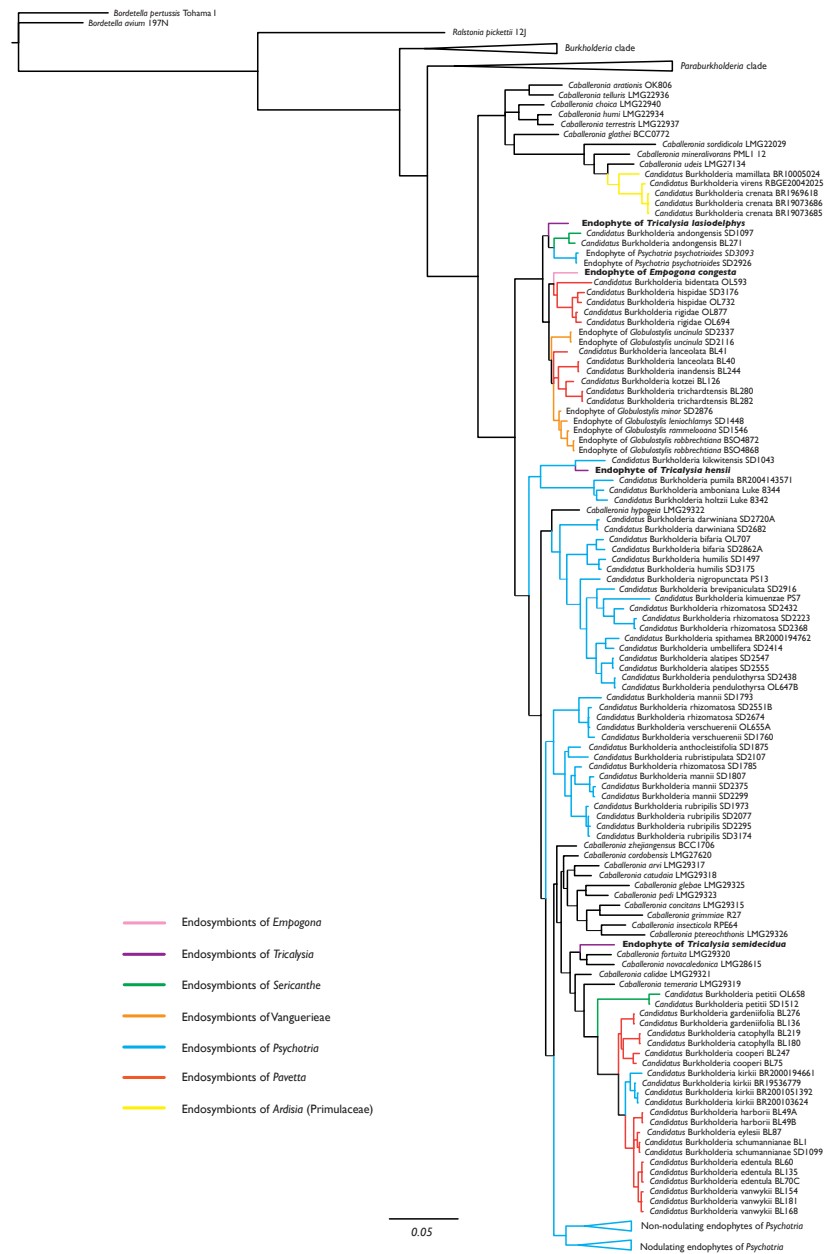

**Figure 1** **Phylogenetic tree of *Burkholderia.* s.l., focussing on *Caballeronia*, based on 16S rRNA, *gyrB*, and *recA* sequences.** The four non-nodulated endophytes in *Empogona* and *Tricalysia* belong to the genus *Caballeronia* and are indicated in bold. Thick lines indicate Bayesian Posterior Probability (BPP) values higher than or equal to 0.95, thin lines indicate BPP support values lower than 0.95.

sequences, *i.e.,* plant DNA sequences in most cases (*e.g.,* *Charr et al., 2020*). For example, in the study of *Ly et al. (2020)*, the objective was to obtain whole chloroplast genomes from 27 species in the Rubiaceae family. Before chloroplast genome assembly, the raw data was "checked in order to detect potential contamination", with the unwanted reads being

removed from further analysis. However, this contamination could be valuable on its own and possibly contain information on the presence of leaf endophytes. In fact, previous studies that used a DNA sequencing approach to detect leaf endophytes in Rubiaceae (*e.g.*, *Van Oevelen et al., 2001*; *Lemaire et al., 2011b*; *Verstraete et al., 2013a*) also extracted total DNA but then eliminated the plant DNA by targeting bacterial DNA.

Our study is based on the unpublished raw data of *Ly et al. (2020)* (but made available here) and looks for evidence of leaf endophytes in the read contamination, specifically focusing on the genera *Empogona* and *Tricalysia*. These two genera belong to the Coffeeae tribe and are closely related to *Sericanthe* (*Arriola et al., 2018*), a genus known for its leaf nodulated symbiosis (*Lemaire et al., 2011a*). Unlike *Sericanthe*, *Empogona* and *Tricalysia* do not have visible nodules in their leaves; the detected leaf endophytes are therefore non-nodulated endophytes. In fact, by examining total DNA, we detected bacterial reads in *E. congesta, T. hensii, T. lasiodelphys*, and *T. semidecidua* (Table 3), indicating the presence of leaf endophytes.

Our reassessment of the original reads of the study of *Ly et al. (2020)* shows that we are dealing with leaf endophytes that are present in large proportions. Epiphytic contamination is unlikely because the leaves were cleaned with sterile water before the extraction of the total DNA. We have found that for each of the four investigated species, a large proportion of the reads is assigned to a limited taxonomic group. In *T. semidecidua,* this even reaches 83% of the named reads. This finding is in line with all previous studies about Rubiaceae endophytes. Although at this point, we cannot rule out that there might be more complex communities within the leaves, the fact remains that the particular *Burkholderia*-Rubiaceae interaction has been demonstrated for this new group of plants.

As such, the fact that leaf endophytes are detected in plants previously not implicated in leaf symbiosis is not unexpected. The wider occurrence of plant-bacteria interactions in Rubiaceae has been suggested before (*Lemaire et al., 2012b*; *Verstraete et al., 2013a*). It is therefore likely that additional hidden plant-bacteria interactions will be found when a systematic survey of leaf symbiosis in Rubiaceae is done.

### The identity of leaf endophytes and their phylogenetic relationships

Our metagenomic analysis revealed that the bacterial leaf endophytes in *Empogona congesta, Tricalysia hensii, T. lasiodelphys*, and *T. semidecidua* belong to the family Burkholderiaceae. This is fully expected as so far, all leaf endophytes in Rubiaceae host plants have been identified as *Burkholderia* s.l. (*e.g.*, *Lemaire et al., 2012b*; *Verstraete, Janssens & Rønsted, 2017*; *Pinto-Carbó et al., 2018*; *Sinnesael, 2020*; *Georgiou et al., 2021*).

After finding out the preliminary identity of the leaf endophytes, three genetic markers were extracted in order to achieve a more accurate identification, as well as to include the newly discovered leaf endophytes in a phylogenetic framework. Analysis of 16S rRNA, *gyrB*, and *recA* has already been extensively used to delineate species within *Burkholderia* s.l., as well as to unravel phylogenetic relationships at the generic level (*Verstraete et al., 2011*; *Verstraete et al., 2013a*; *Lemaire et al., 2011a*; *Lemaire et al., 2011b*). The use of these three markers particularly allows for comparison with other leaf endophytes and *Caballeronia* type strains. Even though the use of genomic data would be preferable and is common in

recent literature about free-living *Burkholderia* s.l. (*Bach et al., 2022*), genomic information about leaf endophytes is often lacking. Such genomic data is usually extracted from pure cultures, but this is not possible for nodulated leaf endophytes, as they cannot be cultivated (*Sinnesael et al., 2019*). The study of the genomes of non-nodulated endophytes shows more promise but this has only just begun (*Danneels et al., 2023*).

The endophytes of *Empogona* and *Tricalysia* are identified as members of the genus *Caballeronia* (Fig. 1). The non-nodulated endophyte of *T. lasiodelphys* is most closely related to nodulated endophytes of *Sericanthe* and non-nodulated endophytes of *Psychotria*, while the endophyte of *E. congesta* is related to nodulated endophytes of *Pavetta* and non-nodulated endophytes of *Globulostylis*. The endophyte of *T. hensii* is most closely related to nodulated and non-nodulated endophytes of *Psychotria*. The endophyte of *T. semidecidua* falls in a clade with *Caballeronia fortuita* and *C. novacaledonica*. The type strain of *C. fortuita* was isolated from *Fadogia homblei* (Rubiaceae) rhizosphere soil in South Africa (*Verstraete et al., 2014*; *Peeters et al., 2016*), while the type strain of *C. novacaledonica* was isolated from *Costularia* (Cyperaceae) rhizosphere soil in New Caledonia (*Guentas et al., 2016*). Besides the fact that all these endophytes belong to the genus *Caballeronia*, there does not seem to be much of a phylogenetic pattern.

The study of *Van Oevelen et al. (2001)*, which was the first to identify leaf endophytes in Rubiaceae host plants (*i.e.*, in a few *Psychotria* species), found that the 16S rRNA sequences of the endophytes were similar to that of *Burkholderia glathei*. As a result, they assigned the Rubiaceae leaf endophytes to the genus *Burkholderia* (*Van Oevelen et al., 2001*). However, since then, several changes have been made to the taxonomy of this genus. First, all leaf endophytes were transferred to *Paraburkholderia*, when *Burkholderia* s.l. was split into a pathogenic group (*Burkholderia* s.s.) and a lineage of environmental bacteria (*Paraburkholderia*) (*Sawana, Adeolu & Gupta, 2014*). Later, *Paraburkholderia* was further subdivided and a new genus was created, *Caballeronia* (*Dobritsa & Samadpour, 2016*), which holds all nodulated endophytes as well as the non-nodulated endophytes of *Globulostylis* and *Psychotria*. The present study also designates the newly discovered non-nodulated endophytes of *Empogona* and *Tricalysia* as species of the genus *Caballeronia* (Fig. 1). The non-nodulated endophytes of the Vanguerieae genera (*Fadogia*, *Fadogiella*, *Rytigynia*, and *Vangueria*) remain in *Paraburkholderia*, except for those of *Globulostylis* (Data S2). This is in agreement with what was previously known (*Verstraete et al., 2013b*).

None of the investigated host plants has conspicuous bacterial leaf nodules in their leaves, and the endophytes are therefore non-nodulated endophytes. When looking at the phylogenetic tree of the leaf endophytes (Fig. 1), there is no apparent phylogenetic pattern for nodulation. The non-nodulated endophytes in *Empogona* and *Tricalysia* are not clustered, although they all belong to the genus *Caballeronia*. The newly found endophytes are related to other nodulated or non-nodulated leaf endophytes. Also, when analysing the results in a larger framework, no phylogenetic pattern is apparent for all non-nodulated endophytes in Rubiaceae since the majority of the Vanguerieae endophytes are situated within the genus *Paraburkholderia* (Data S2). However, we hypothesize that leaf nodulation should be considered as a character of the host plants, rather than of the leaf endophytes (see also *Lemaire et al., 2012b*).

## Patterns in the host plants

In this study, we found *Burkholderia* s.l. endophytes in two genera of Rubiaceae that were previously not known to take part in leaf symbiosis. This brings the total number of genera in Rubiaceae for which leaf symbiosis is (molecularly) confirmed to ten: *Psychotria* (Psychotrieae tribe), *Pavetta* (Pavetteae tribe), *Fadogia*, *Fadogiella*, *Globulostylis*, *Rytigynia*, *Vangueria* (Vanguerieae tribe), and *Empogona*, *Sericanthe*, *Tricalysia* (Coffeeae tribe).

Finding phylogenetic patterns is however challenging. For the five Vanguerieae genera, the presence of *Burkholderia* s.l. endophytes is consistent at the genus level (*Verstraete et al., 2013a*) and the plants with leaf symbiosis only occur in Africa and Madagascar. For the genus *Pavetta*, it is generally assumed that most of the species have leaf nodules (*Lersten & Horner, 1976*; *Miller, 1990*; *Lemaire et al., 2011b*) and these are found in the Paleotropics (*POWO, 2023*). The presence or absence of nodules, as well as their form, has been used in the past to classify subgeneric taxa (*e.g.*, *P.* series *Enodulosae*; *Bremekamp, 1939*). However, within *Pavetta*, the phylogenetic distribution of species without nodules is irregular (*Bremekamp, 1934*). Within the pantropical genus *Psychotria*, the situation is even more complex. The number of (known) species with nodules (ca 80 spp; *Lemaire et al., 2012b*) is rather limited compared to the total number of species in the genus (ca 1645; *POWO, 2023*, meaning that the nodulated form of leaf symbiosis is not a frequent character in *Psychotria*. Also, nodulating *Psychotria* plants are restricted to Africa and Madagascar. Unfortunately, a detailed list with the presence and absence of nodules is missing, so it remains uncertain to date to what extent leaf nodulation is present in *Psychotria*. The few nodulating *Psychotria* species that were included in molecular studies were not recovered as a monophyletic group (*Razafimandimbison et al., 2014*). Besides nodulating species, there are also some species without nodules but with non-nodulated endophytes (*Lemaire et al., 2012b*). Although these non-nodulating *Psychotria* plants were found in a clade (clade III in *Lemaire et al., 2012b*) separate from the nodulating species (clade II in *Lemaire et al., 2012b*), they also do not form a monophyletic group. Finally, the species *Psychotria lucens* Hiern was used in the past as a negative control (*Van Oevelen et al., 2001*) and later several other species without bacterial endophytes were found (*Lemaire et al., 2012b*). This means that all three conditions occur in *Psychotria*. However, the taxonomic delimitation of *Psychotria* has changed many times (*Razafimandimbison et al., 2014*) and finding evolutionary patterns within this megagenus remains challenging.

The genera *Empogona*, *Sericanthe*, and *Tricalysia* are closely related (*Arriola et al., 2018*), and the species of *Empogona* (*Tosh et al., 2009*) and *Sericanthe* (*Robbrecht, 1978*) were once included in *Tricalysia*. Perhaps it is therefore not so surprising to find leaf endophytes in these genera. The difference between *Sericanthe* on the one hand and *Empogona* and *Tricalysia* on the other, is that the former has leaf nodules, while the latter do not. A next step would be to investigate additional species of *Empogona* and *Tricalysia* to elucidate the true extent of leaf symbiosis in these two genera and to find out whether leaf symbiosis has any phylogenetic signal. Another observation worth investigating is that all tree genera are related to *Diplospora* (*Arriola et al., 2018*) and *Discospermum* Dalzell (*Tosh et al., 2009*). However, the major difference is that these two genera are found in (sub)tropical Asia, while the other three genera (*Empogona*, *Sericanthe*, and *Tricalysia*) are exclusively found

in continental Africa and Madagascar (*POWO, 2023*). So far, the nodulated symbiosis is predominantly found in Africa (except for a few noduled *Pavetta* species in (sub)tropical Asia) and non-nodulated symbiosis is even restricted to that area. A broader screening of the Coffeeae tribe would therefore be useful to demonstrate the presence or absence of a geographic pattern in leaf symbiosis. However, for this, a new phylogenetic framework of the Coffeeae is needed, which shows the relationships between the different genera and onto which the character "leaf symbiosis" can then be plotted.

## CONCLUSIONS

Metagenomic analysis revealed that bacterial endophytes are present in the leaves of *Empogona* and *Tricalysia*, two genera not previously implicated in leaf symbiosis. This result is another step towards discovering the true extent of leaf symbiosis (both nodulated and non-nodulated) in the Rubiaceae family. The endophytes belong to the genus *Caballeronia* and are not housed in leaf nodules. No phylogenetic signals have been found in the endophytes, nor does there appear to be a phylogenetic or geographical pattern in the host species. However, leaf symbiosis is predominantly found in Africa (as are both *Empogona* and *Tricalysia*), so additional plant-bacteria interactions are likely to be found on this continent.

## ACKNOWLEDGEMENTS

The authors thank Dr Mathilde Dupeyron for her input on the manuscript. The IFB Core Cluster that is part of the National Network of Computing Resources (NNCR) of the Institut Francais de Bioinformatique provided HPC resources.

### Funding

The authors received no funding for this work.

### Competing Interests

The authors declare there are no competing interests.

### Author Contributions

- Brecht Verstraete conceived and designed the experiments, performed the experiments, analyzed the data, prepared figures and/or tables, authored or reviewed drafts of the article, and approved the final draft.
- Steven Janssens conceived and designed the experiments, performed the experiments, analyzed the data, prepared figures and/or tables, authored or reviewed drafts of the article, and approved the final draft.
- Petra De Block conceived and designed the experiments, authored or reviewed drafts of the article, and approved the final draft.
- Pieter Asselman performed the experiments, prepared figures and/or tables, authored or reviewed drafts of the article, and approved the final draft.

- Gabriela Méndez performed the experiments, analyzed the data, prepared figures and/or tables, authored or reviewed drafts of the article, and approved the final draft.
- Serigne Ly performed the experiments, analyzed the data, authored or reviewed drafts of the article, and approved the final draft.
- Perla Hamon conceived and designed the experiments, authored or reviewed drafts of the article, and approved the final draft.
- Romain Guyot conceived and designed the experiments, performed the experiments, analyzed the data, prepared figures and/or tables, authored or reviewed drafts of the article, and approved the final draft.

## DNA Deposition

The following information was supplied regarding the deposition of DNA sequences:

The raw Illumina reads are available at GenBank: PRJNA880288, as well as for each species separately (SRR21547707, SRR21547708, SRR21547709, SRR21547710, Table 1).

The draft genome assemblies of the four endophytes using MaSuRCa are available at GenBank: JAQFVG000000000, JAQFVH000000000, JAQFVI000000000, JAQFVJ000000000 (Table 3) and Zenodo: Verstraete, Brecht, Janssens, Steven, de Block, Petra, Asselman, Pieter, Mendez Silva, Gabriela, Ly, Serigne Ndiawar, Hamon, Perla, & Guyot, Romain. (2022). Metagenomics of African *Empogona* and *Tricalysia* (Rubiaceae) reveals the presence of leaf endophytes. https://doi.org/10.5281/zenodo.6090258.

The draft genome assemblies of the four endophytes using metaSPAdes is available on Zenodo: Verstraete, Brecht, Janssens, Steven, De Block, Petra, Asselman, Pieter, Mendez Silva, Gabriela, Ndiawar Ly, Serigne, Hamon, Perla, & Guyot, Romain. (2023). Metagenomics of African *Empogona* and *Tricalysia* (Rubiaceae) reveals the presence of leaf endophytes-Assembly and annotation Files. https://doi.org/10.5281/zenodo.7787854.

The 16 rRNA, *gyrB*, and *recA* sequences of the four endophytes are available at GenBank: OQ189898, OQ189899, OQ189900, OQ189901, OQ305553, OQ305554, OQ305555, OQ305556, OQ305557, OQ305558 (Table S1) and at Zenodo: Verstraete, Brecht, Janssens, Steven, De Block, Petra, Asselman, Pieter, Mendez Silva, Gabriela, Ndiawar Ly, Serigne, Hamon, Perla, & Guyot, Romain. (2022). Metagenomics of African *Empogona* and *Tricalysia* (Rubiaceae) reveals the presence of leaf endophytes-Fasta files. https://doi.org/10.5281/zenodo.7333199

The Kaiju output is available in Data S1. The data used in the phylogenetic analysis is available in Table S1. The full phylogenetic tree of *Burkholderia* s.l. is available in Data S2.

## Data Availability

The raw Illumina reads are available at GenBank: PRJNA880288, as well as for each species separately (SRR21547707, SRR21547708, SRR21547709, SRR21547710, Table 1).

The draft genome assemblies of the four endophytes using MaSuRCa are available at GenBank: JAQFVG000000000, JAQFVH000000000, JAQFVI000000000, JAQFVJ000000000 (Table 3) and Zenodo: Verstraete, Brecht, Janssens, Steven, de Block, Petra, Asselman, Pieter, Mendez Silva, Gabriela, Ly, Serigne Ndiawar, Hamon, Perla, &

Guyot, Romain. (2022). Metagenomics of African *Empogona* and *Tricalysia* (Rubiaceae) reveals the presence of leaf endophytes. https://doi.org/10.5281/zenodo.6090258.

The draft genome assemblies of the four endophytes using metaSPAdes is available on Zenodo: Verstraete, Brecht, Janssens, Steven, De Block, Petra, Asselman, Pieter, Mendez Silva, Gabriela, Ndiawar Ly, Serigne, Hamon, Perla, & Guyot, Romain. (2023). Metagenomics of African *Empogona* and *Tricalysia* (Rubiaceae) reveals the presence of leaf endophytes-Assembly and annotation Files. https://doi.org/10.5281/zenodo.7787854.

The 16 rRNA, *gyrB*, and *recA* sequences of the four endophytes are available at GenBank: OQ189898, OQ189899, OQ189900, OQ189901, OQ305553, OQ305554, OQ305555, OQ305556, OQ305557, OQ305558 (Table S1) and at Zenodo: Verstraete, Brecht, Janssens, Steven, De Block, Petra, Asselman, Pieter, Mendez Silva, Gabriela, Ndiawar Ly, Serigne, Hamon, Perla, & Guyot, Romain. (2022). Metagenomics of African *Empogona* and *Tricalysia* (Rubiaceae) reveals the presence of leaf endophytes-Fasta files. https://doi.org/10.5281/zenodo.7333199

The Kaiju output is available in Data S1. The data used in the phylogenetic analysis is available in Table S1. The full phylogenetic tree of *Burkholderia* s.l. is available in Data S2.

## Supplemental Information

Supplemental information for this article can be found online at http://dx.doi.org/10.7717/peerj.15778#supplemental-information.

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
