# Peer review of "Metagenomics of African Empogona and Tricalysia (Rubiaceae) reveals the presence of leaf endophytes"

_PeerJ, doi:10.7717/peerj.15778_

## Round 0.1 · original submission · Major Revisions

Dear authors,

Two reviewers are generally positive. But they still recommend some changes, especially concerning the methods as indicated by rev. #1.
Please address all their points.
Best wishes
Mike Thiv

·

Basic reporting

According to the manuscript “Our study is based on the data of Ly et al. (2020)” (L264, L272), but paper Ly, 2020 contains no raw data published.
L287: There are more recent studies on the topic to be cited:
• Georgiou, A., Sieber, S., Hsiao, CC. et al. Leaf nodule endosymbiotic Burkholderia confer targeted allelopathy to their Psychotria hosts. Sci Rep 11, 22465 (2021). https://doi.org/10.1038/s41598-021-01867-2
• Bacterial leaf symbiosis–Origin, function, evolutionary gain, and transmission mode of endophytes in bacteriophilous Rubiaceae (https://lirias.kuleuven.be/2909132?limo=0)
Thick and thin lines on Fig. 2 are hard to distinguish, I suggest to use dashed or colored lines. Text on Fig. 1 is unreadable, the report archives from Kaiju better be published in Supplementary materials or at Figshare/Zenodo.

Experimental design

The assembly software choice is arbitrary. I recommend to write the criteria and requirements for the assembler based on data. The reads quality filtering procedure details are missing (L134). Some review articles may help to form these criteria: https://doi.org/10.1016/j.mimet.2018.06.007 , https://www.mdpi.com/2076-2607/10/12/2416
Which reads were discarded after FASTQC analysis?
Which particular database type and version were used for Kaiju to (L130) “to identify and classify reads not belonging to the plant genome with the NCBI non-redundant RefSeq protein database”? There are “nr_euk”, “nr” and “refseq” types fits the description above.
Authors used some outdated software: BUSCO v.2.0 was released in 2019, Kaiju 1.8.2 in 2021.

Validity of the findings

If three of four assembled genomes have high BUSCO score (Table 3), why not to annotate and publish them to the NCBI GenBank? The Venn diagram for these annotated genomes might show the common and different genes between close bacterial species.
To extract 16S rRNA genes from metagenome data MATAM software could be used, it contains the reference database based on SILVA.
I performed reads classification for SRR21547707 and SRR21547709 with Kraken2 and database PlusPFP database (https://benlangmead.github.io/aws-indexes/k2). The percentage of reads belong to particular organisms is follows:
SRR21547707 (T. semidecidua): 31.56 – Bacteria; 25.45 – Burkholderiales
SRR21547709 (T. hensii): 14.89 – Bacteria; 10.59 – Burkholderiales
Theses % fits with authors’ results, but PlusPFP database allows to evaluate each read among all kingdoms of life. As the genomes of endophytic bacteria are assembled, they might be used to construct the phylogenic tree using PhyloPhlAn.

Reviewer 2 ·

Basic reporting

The manuscript by Verstraete et al. entitled « Metagenomics of African Empogona and Tricalyisia (Rubiaceae) reveals the presence of leaf endophytes » describes the detection of putative leaf symbionts of the genus Burkholderia s.l. in leaf samples of several Rubiaceae plants. The authors cleverly use previously generated data to expand the range of known leaf symbioses in the Rubiaceae, adding 2 genera belonging to the Coffeeae tribe.
The manuscript is well written, and the main conclusion that endophytic Burkholderia were detected in the four datasets examined is well supported by the evidence. Aside from minor issues with legends (see "additional comments"), figures, tables, and citations are up to a professional standard.

The sequence data are deposited in a general-purpose repository (Zenodo), but should be deposited in a sequence repository such as Genbank. I truly appreciate the authors making their data available through a public repository, but I believe that this is insufficient: Sequence data on Zenodo are not included in any of the major sequence databases, are not searchable, and do not include standardized metadata. Given the importance of this study in describing new leaf symbioses, I suggest submitting their assemblies to a public sequence repository.

Experimental design

The highly fragmented nature of the assembled genomes and low BUSCO scores indicate that MAGs may be incomplete. This is not completely unusual for metagenome assemblies, but the fact that the authors chose to select reads based on nucleotide composition and coverage prior to assembly could introduce bias. This would result in incomplete bins, as the composition and kmer coverage of individual (bacterial) reads may not be homogenous enough to pass the filter.
It seems also that contigs were selected for analysis based on similarity to Burkholderia s.l. genomes, discarding contigs that do not yield significant hits. This might create some taxonomic bias, masking the presence of genome information not represented in the database (e.g. other taxa or genes acquired via horizontal transfer).
Instead, the standard procedure in the field is to assemble reads with software adapted to metagenome assembly (e.g. MetaSPAdes or similar), and bin the contigs according to coverage and nucleotide composition. Nearly fully automated pipelines are available that perform metagenome assembly and binning (e.g. nfcore/mag, AutoMETA and many others). Resulting MAGs, once evaluated for completeness and contamination, may be used to get a more accurate genome size, and consequently a better picture of endophyte diversity and abundance in the samples. Better MAG accuracy would allow genome comparison with Caballeronia type species and available genomes of other leaf symbionts. Crucially, this would help determine the make-up of endophytic communities, i.e. whether only one taxon or several taxa are present.

Validity of the findings

The conclusion that Burkholderia s.l. are detected in the samples is well demonstrated, but because of methodological shortcomings (see comments in "Experimental design"), it is not clear whether these taxa occur alone or in complex communities within leaves.

Moreover, I feel that the authors do not fully exploit the potential of their data to shed light on Burkholderia leaf symbiosis in these new taxa. In particular, taxonomic analysis based on comparison with the genomes of Caballeronia type strains should be undertaken to clarify the taxonomic status of the endophytic bacteria.
Although the whole genome of the 4 potential symbionts are assembled, I find it a pity that no attempt is made at annotating the genomes. This would allow the reporting of basic properties such as number of genes, repeats and maybe some functional analysis. I understand that this type of analysis may not be within the immediate scope of the study, but this is fairly easy to do nowadays with automated, publicly available pipelines and I see little reason not to provide this information to the reader.

Additional comments

I have a few additional minor comments listed below :

L68 : « Every plant species seems to have its own unique bacterial species ». This is not necessarily true (see Danneels et al 2022), please rephrase to something along the lines of « most plant species harbor unique bacterial lineages ».

L129. Was the kaiju classification done on raw reads, or reads that had beed already labeled as « contaminants » ?

L140 : The GC content values given most probably are counts per 27-mer and not %. Please correct if this is the case.

L145. If I understand this correctly, contigs were filtered based on similarity with Burkholderia s.l. genomes, discarding contigs that do not yield significant hits. This might create some taxonomic bias, masking the presence of taxa not represented in the database.

L172 : Please explain what you mean by « possible incongruence between datasets ».
Table 2 : Please include average coverage per genome assembly.

L195, 197, 199. I suggest changing the phrasing « be assigned a taxon name » to « assigned to a bacterial taxon », to be less ambiguous.

L208. GC content of 15-20% strikes me as very low for Burkholderia s.l. (usually > 60%). The authors possibly confuse the « GC count » of the KAT tool, which is the number of G or C in 27-mers, with GC content. Please update the text as appropriate throughout the paragraph.

L214. These are rather low N50 values, which may suggest incomplete read sets, the presence of polymorphisms or repeats. Most likely, the strategy used by the authors is responsible to some extent. Filtering reads before assembly is likely to result in gaps in the assembly. The criteria used for filtering the reads are not clear to me : Were reads that contained at least 1 kmer below the threshold value entirely discarded ?

Table 3. Except for the genome assembled from T. hensii, the BUSCO scores are low to very low, indicating that the genomes are not complete. See comment about assembly strategy.

Fig1. I believe the Y-axis on panel B represents GC count per kmer, not GC content as it is commonly interpreted. I suggest updating the legend to avoid confusion.

Figure 2. Type strains of Caballeronia species should be included to place the endophytes relative to validly described taxa.

Figure 2. The legend indicates that thick and thin lines represent different branch support thresholds, but on my screen all lines seem to have the same thickness. It is then impossible to say if branch support is high or low.

---

## Round 0.2 · accepted · Accept

Dear authors,
Your manuscript is now accepted. Congrats!
Best
Mike Thiv

Reviewer 2 ·

Basic reporting

The authors have adequately answered my comments on the previous version of the manuscript and deposited their sequence data in public repositories.

Experimental design

The authors properly acknowledge the limitation of their approach (read filtering followed by assembly) in the revised manuscript. Although more sophisticated methods are available which may yield better results, their limited access to high performance computing infrastructure prevented them from doing so. I am nevertheless satisfied that the main claim of the paper is well supported, and that the reader is sufficiently made aware of potential pitfalls.

Validity of the findings

The authors adequately answered my comments on the previous version of the manuscript.

Additional comments

The revised manuscript is clearer and the authors took great care in answering the major criticisms from the reviewers. I have no additional comment.

---

## Author Rebuttal · Round 0.2

**Response to the reviewers for the manuscript "Metagenomics of African *Empogona* and *Tricalysia* (Rubiaceae) reveals the presence of leaf endophytes" submitted to PeerJ**

Reviewer 1

**Basic reporting**

*According to the manuscript "Our study is based on the data of Ly et al. (2020)" (L264, L272), but paper Ly, 2020 contains no raw data published.*
- ✓ The raw data of Ly et al. (2020) was indeed not released at the time, but our study is based on the observations from 2020. The data for the four *Empogona/Tricalysia* species is here released at NCBI with the following accession numbers: SRR21547707, SRR21547708, SRR21547709, and SRR21547710 (new Table 1). We changed the sentence in the text to indicate this (new L271–L273).

*L287: There are more recent studies on the topic to be cited:*
*• Georgiou, A., Sieber, S., Hsiao, CC. et al. Leaf nodule endosymbiotic Burkholderia confer targeted allelopathy to their Psychotria hosts. Sci Rep 11, 22465 (2021). https://doi.org/10.1038/s41598-021-01867-2*
*• Bacterial leaf symbiosis–Origin, function, evolutionary gain, and transmission mode of endophytes in bacteriophilous Rubiaceae (https://lirias.kuleuven.be/2909132?limo=0)*
- ✓ Both references were added (new L299), although we want to point out that the second reference is a PhD thesis that is currently not freely accessible and its chapters consist of separately published freely accessible papers, like Sinnesael et al. 2019, which was already cited.

*Thick and thin lines on Fig. 2 are hard to distinguish, I suggest to use dashed or colored lines.*
- ✓ This figure has been changed.

*Text on Fig. 1 is unreadable, the report archives from Kaiju better be published in Supplementary materials or at Figshare/Zenodo.*
- ✓ Figure 1 has been merged with the new Figure S1 and the Kaiju output is provided as Supplemental Data S1.

**Experimental design**

*The assembly software choice is arbitrary. I recommend to write the criteria and requirements for the assembler based on data.*
- ✓ According to our experience with eukaryote genome assemblies, MaSuRCa gives better results than other "short read" assemblers, particularly with hybrid assembly (https://f1000research.com/posters/9-764; https://doi.org/10.1186/s12864-020-07041-8). We used the default parameters of MaSuRCa, since it selects itself the best parameters (k-mer length, genome size, correction) for assembly depending on the data properties. However, following this comment, we used an alternative assembler, metaSPAdes (new L144–L146), and the results are provided on Zenodo (https://doi.org/10.5281/zenodo.7787854).

*The reads quality filtering procedure details are missing (L134). Some review articles may help to form these criteria: https://doi.org/10.1016/j.mimet.2018.06.007, https://www.mdpi.com/2076-2607/10/12/2416.*

- ✓ Although the quality has been explored, we did not apply a filter before the assembly step because MaSuRCa uses its own criteria. We just checked that no adaptations remained in datasets. In the MaSuRCa manual is says: "IMPORTANT! Avoid using third party tools to pre-process the Illumina data before providing it to MaSuRCa, unless you are absolutely sure you know exactly what the preprocessing tool does. Do not do any trimming, cleaning or error correction. This will likely deteriorate the assembly." (https://github.com/alekseyzimin/masurca). We added some information in the text to help the reader understand this step (new L141–L142).

*Which reads were discarded after FASTQC analysis?*

- ✓ No reads were discarded after FASTQC analysis. FASTQC was used to check for "contamination" (new L133–L135).

*Which particular database type and version were used for Kaiju to (L130) "to identify and classify reads not belonging to the plant genome with the NCBI non-redundant RefSeq protein database"? There are "nr_euk", "nr" and "refseq" types fits the description above.*

- ✓ We used the NCBI nr_euk. This information was added to the text (new L131–L132).

*Authors used some outdated software: BUSCO v.2.0 was released in 2019, Kaiju 1.8.2 in 2021.*

- ✓ We used Kaiju v.1.9.0 and reran BUSCO v.5.4.3 with odb10 (new L130 and L153–L155).

**Validity of the findings**

*If three of four assembled genomes have high BUSCO score (Table 3), why not to annotate and publish them to the NCBI GenBank? The Venn diagram for these annotated genomes might show the common and different genes between close bacterial species.*

- ✓ We would like to say yes, but there are several considerations to take into account. The produced genomes are probably incomplete and fragmented ("draft") due to the filtering process (k-mer/GC count + BLAST) and so, in this context, it is not advisable to produce an annotation and release it to NCBI. Secondly, the objective of this article is not to analyse and compare the genomes, although it might seem an interesting idea, but to show the presence of proteobacteria in the plants and to use assembly for phylogenetic analysis. In the future, we would probably use long read sequencing to sequence these genomes and to produce a high-quality assembly for comparative genomic analysis. However, we acknowledge the comment by the reviewer and produced annotations using Prokka (new L155–L156) and released them on Zenodo (https://doi.org/10.5281/zenodo.7787854).

*To extract 16S rRNA genes from metagenome data MATAM software could be used, it contains the reference database based on SILVA. I performed reads classification for SRR21547707 and SRR21547709 with Kraken2 and database PlusPFP database (https://benlangmead.github.io/aws-indexes/k2). The percentage of reads belong to particular organisms is as follows:*
*SRR21547707 (T. semidecidua): 31.56 – Bacteria; 25.45 – Burkholderiales*
*SRR21547709 (T. hensii): 14.89 – Bacteria; 10.59 – Burkholderiales*
*Theses % fits with authors' results, but PlusPFP database allows to evaluate each read among all kingdoms of life. As the genomes of endophytic bacteria are assembled, they might be used to construct the phylogenetic tree using PhyloPhlAn.*

✓ Thank you for your kind advice. Since we do not consider ourselves as part of the metagenomic research field, we developed our own pipeline to identify and extract specific genes from the assemblies. We tried MATAM and got the same results, which strengthens our confidence in our results obtained with our pipeline (new L159–L160).

Reviewer 2

**Basic reporting**

*The sequence data are deposited in a general-purpose repository (Zenodo), but should be deposited in a sequence repository such as Genbank. I truly appreciate the authors making their data available through a public repository, but I believe that this is insufficient: Sequence data on Zenodo are not included in any of the major sequence databases, are not searchable, and do not include standardized metadata. Given the importance of this study in describing new leaf symbioses, I suggest submitting their assemblies to a public sequence repository.*

✓ We agree and we submitted the raw data (SRR21547707, SRR21547708, SRR21547709, SRR21547710; new Table 1) and the assemblies (JAQFVG000000000, JAQFVH000000000, JAQFVI000000000, JAQFVJ000000000; new Table 3) to NCBI.

**Experimental design**

*The highly fragmented nature of the assembled genomes and low BUSCO scores indicate that MAGs may be incomplete. This is not completely unusual for metagenome assemblies, but the fact that the authors chose to select reads based on nucleotide composition and coverage prior to assembly could introduce bias. This would result in incomplete bins, as the composition and kmer coverage of individual (bacterial) reads may not be homogenous enough to pass the filter. It seems also that contigs were selected for analysis based on similarity to Burkholderia s.l. genomes, discarding contigs that do not yield significant hits. This might create some taxonomic bias, masking the presence of genome information not represented in the database (e.g. other taxa or genes acquired via horizontal transfer).*

✓ We completely agree with this comment. In order to properly separate the data between plant DNA sequences and proteobacteria sequences, we applied a k-mer/GC count filter, which could lead to bias. In addition, we also used a BLAST filter to be sure to remove any remaining plant/bacteria sequences. The goal was

to avoid submitting a bad genome sequence to NCBI. We have added some additional information in the text to alert the reader of this bias (new L232–234).

*Instead, the standard procedure in the field is to assemble reads with software adapted to metagenome assembly (e.g. MetaSPAdes or similar), and bin the contigs according to coverage and nucleotide composition. Nearly fully automated pipelines are available that perform metagenome assembly and binning (e.g. nfcore/mag, AutoMETA and many others). Resulting MAGs, once evaluated for completeness and contamination, may be used to get a more accurate genome size, and consequently a better picture of endophyte diversity and abundance in the samples. Better MAG accuracy would allow genome comparison with Caballeronia type species and available genomes of other leaf symbionts. Crucially, this would help determine the make-up of endophytic communities, i.e. whether only one taxon or several taxa are present.*

✓ Thank you for kind advice. We ran metaSPAdes on k-mer/GC filtered reads, see below for the results. Although it sometimes improved the N50 and/or BUSCO, metaSPAdes failed to run on raw Illumina data due to RAM memory errors. It failed with 144 GB RAM and it will probably require more time to bypass this computational issue.

| | Contig N50 (bp) | Assembly size (Mb) | n (> 500 bp) | Complete Busco | Missing Busco |
|---|---|---|---|---|---|
| *Empogona congesta (metaspades)* | 33610 | 4.090 | 323 | 97.8% | 1.3% |
| *Empogona congesta (Masurca + Blast filter)* | 10140 | 3.762 | 612 | 93.2% | 1.8% |
| *Tricalysia hensii (metaspades)* | 54817 | 7.905 | 709 | 99.6% | 0.4% |
| *Tricalysia hensii (Masurca + Blast filter)* | 60447 | 6.951 | 369 | 99.6% | 0.4% |
| *Tricalysia lasiodelphys (metaspades)* | 9304 | 4.283 | 1197 | 93.2% | 2.2% |
| *Tricalysia lasiodelphys (Masurka + Blast filter)* | 2165 | 4.145 | 2644 | 75.3% | 6.4% |
| *Tricalysia semidecidua (metaspades)* | 32801 | 4.441 | 440 | 98.7% | 0.8% |
| *Tricalysia semidecidua (Masurka + Blast filter)* | 24168 | 3.919 | 490 | 95.4% | 1.4% |

**Validity of the findings**

*The conclusion that Burkholderia s.l. are detected in the samples is well demonstrated, but because of methodological shortcomings (see comments in "Experimental design"), it is not clear whether these taxa occur alone or in complex communities within leaves.*

✓ An expanded discussion regarding this topic has been included (new L279–L287).

*Moreover, I feel that the authors do not fully exploit the potential of their data to shed light on Burkholderia leaf symbiosis in these new taxa. In particular, taxonomic analysis based on comparison with the genomes of Caballeronia type strains should be undertaken to clarify the taxonomic status of the endophytic bacteria.*

✓ The *Caballeronia* type strains have been added to the phylogenetic analysis and their metadata is added to Table S1. The new Figure 1 now includes the type

strains but the general topology of the tree has not changed. Also, the position and the relationships of the new endophytes is identical as well. However, the relationship of the endophyte of *T. semidecidua* is now clearer: it is related to *Caballeronia fortuita* and *C. novacaledonica*, two rhizosphere bacteria. The Materials & Methods (new L174–L176), Results (new L241–L249), and Discussion (new L313–L324) have been adapted to reflect the changes.

*Although the whole genome of the 4 potential symbionts are assembled, I find it a pity that no attempt is made at annotating the genomes. This would allow the reporting of basic properties such as number of genes, repeats and maybe some functional analysis. I understand that this type of analysis may not be within the immediate scope of the study, but this is fairly easy to do nowadays with automated, publicly available pipelines and I see little reason not to provide this information to the reader.*
- ✓ We produced annotations using Prokka (new L155–L156) and released them on Zenodo (https://doi.org/10.5281/zenodo.7787854).

**Additional comments**

*L68. « Every plant species seems to have its own unique bacterial species ». This is not necessarily true (see Danneels et al 2022), please rephrase to something along the lines of « most plant species harbor unique bacterial lineages ».*
- ✓ The sentence has been rephrased (new L68–L69).

*L129. Was the kaiju classification done on raw reads, or reads that had been already labeled as « contaminants »?*
- ✓ The Kaiju classification was done on the raw reads to check for contamination before processing the reads for assembly of the plant genomes (new L129–L132).

*L140. The GC content values given most probably are counts per 27-mer and not %. Please correct if this is the case.*
- ✓ The sentence has been rephrased (new L137–L139).

*L145. If I understand this correctly, contigs were filtered based on similarity with Burkholderia s.l. genomes, discarding contigs that do not yield significant hits. This might create some taxonomic bias, masking the presence of taxa not represented in the database.*
- ✓ We agree. The objective of this part was not to reassemble the best genomes, but to avoid sequences of plants or other organisms in our assembly and to identify targeted genes for phylogenetic analysis.

*L172. Please explain what you mean by « possible incongruence between datasets ».*
- ✓ This is default test used in phylogenetics when combining multiple sources of phylogenetic data (i.e. several independent DNA markers). It tests the effect of the combined dataset on phylogenetic accuracy to avoid biased estimates when a substitution model and its parameters are a poor fit to one of the partitions.

*Table 2: Please include average coverage per genome assembly.*
- ✓ Average coverage was added to the new table 2.

*L195, 197, 199. I suggest changing the phrasing « be assigned a taxon name » to « assigned to a bacterial taxon », to be less ambiguous.*
  - ✓ "Bacterial taxon" would not be entirely correct since the Kaiju output also contains a tiny fraction of non-bacterial names.

*L208. GC content of 15-20% strikes me as very low for Burkholderia s.l. (usually > 60%). The authors possibly confuse the « GC count » of the KAT tool, which is the number of G or C in 27-mers, with GC content. Please update the text as appropriate throughout the paragraph.*
  - ✓ Suggestion accepted for several instances (new L194–L216).

*L214. These are rather low N50 values, which may suggest incomplete read sets, the presence of polymorphisms or repeats. Most likely, the strategy used by the authors is responsible to some extent. Filtering reads before assembly is likely to result in gaps in the assembly. The criteria used for filtering the reads are not clear to me: were reads that contained at least 1 kmer below the threshold value entirely discarded?*
  - ✓ We indicated in the text that we consider the assembly as a rough draft. The reads that contained 1 k-mer below the threshold were discarded, as well as its pair (new L232–L234).

*Table 3. Except for the genome assembled from T. hensii, the BUSCO scores are low to very low, indicating that the genomes are not complete. See comment about assembly strategy.*
  - ✓ See our reply to the comment about the assembly strategy above.

*Figure 1. I believe the Y-axis on panel B represents GC count per kmer, not GC content as it is commonly interpreted. I suggest updating the legend to avoid confusion.*
  - ✓ The legend has been updated accordingly.

*Figure 2. Type strains of Caballeronia species should be included to place the endophytes relative to validly described taxa.*
  - ✓ See our reply to the comment about comparing with the type strains above.

*Figure 2. The legend indicates that thick and thin lines represent different branch support thresholds, but on my screen all lines seem to have the same thickness. It is then impossible to say if branch support is high or low.*
  - ✓ This figure has been changed.